# Ultrasound-Based Real-Time Imaging of Hydrogel-Based Millirobots with Volume Change Capability

**DOI:** 10.3390/mi14020422

**Published:** 2023-02-10

**Authors:** Yaxin Hou, Yuguo Dai, Wei Zhang, Minghui Wang, Hanxue Zhao, Lin Feng

**Affiliations:** 1Department of Diagnostic Ultrasound, Beijing Tongren Hospital, Capital Medical University, Beijing 100730, China; 2Department of Mechanical Engineering, The University of Tokyo, Tokyo 113-8656, Japan; 3Department of Materials Science and Engineering, College of Design and Engineering, National University of Singapore, Singapore 117583, Singapore; 4NO. 1 Outpatient Department of Xi Cheng District, Beijing Garrison, Beijing 100055, China; 5School of Mechanical Engineering and Automation, Beihang University, Beijing 100191, China; 6Beijing Advanced Innovation Center for Biomedical Engineering, Beihang University, Beijing 100083, China

**Keywords:** millirobot, ultrasound imaging, magnetic control, NIR irradiation

## Abstract

Soft-bodied robots driven by external fields have better environmental adaptability, extending their applications. Nature also provides lots of inspiration for shape-morphing robot development, for example, larvae and jellyfish. This paper presents magnetically propelled hydrogel-based millirobots with volume changeability. The millirobot can be imaged in real time in a completely enclosed space with an ultrasound imaging system. Firstly, a custom-designed magnetic generating system with six square coils was introduced to generate a uniform field to propel the robot. The robot was fabricated using hydrogel with a thickness of around 300 μm. After programmable magnetization, the robot could change its shape and move using the rotating magnetic field. With the near-infrared illumination, the robot could shrink and could recover when the illumination stopped. Even when the robot shrank, it could be propelled by the external field, showing its potential usage in complex environments. Moreover, the posture information of the robot including the position and shape could be obtained in real time using ultrasound image technology.

## 1. Introduction

Recent years have seen a proliferation of robots in both engineering and biomedical applications [1,2,3]. However, conventional rigid miniature robots have limited locomotor performance due to their inability to cope with obstacles and texture changes in unstructured environments. Soft-bodied robots, on the other hand, possess unique characteristics such as shape deformation and changes in rigidity, making them well suited to operating in challenging conditions [4,5]. Additionally, these robots can access confined spaces and be controlled in a wireless and noninvasive manner [6,7,8,9]. Propelled by external fields, such as a magnetic field [10,11,12,13], electrical field [14], pressure [15], etc., specially designed robots constructed from soft materials could exhibit versatile morphologies and interact with the environment, leading to their locomotion. Among these robots, magnetic soft robots that are capable of fast responsiveness, reversible shape transformation ability, etc., are widely used in biomedical devices [16], and they have great application potential in drug delivery and object manipulation.

Shape-morphing robots that can respond to external magnetic fields rapidly with time-varying shapes are typically constructed from magnetic materials with low elastic modulus [17,18,19]. The shape-morphing capacity is determined by the shape and magnetization profile of the robot. Therefore, designing robots with special magnetization profiles is crucial since this determines the way the robots interact with the environments. Typically, an auxiliary tool such as a mold is used to program the magnetization profile in a strong magnetic field [20]. Once the robot is specially magnetized, magnetic forces or torques are applied to the robot to align the magnetic moment with the external field. By applying a time-dependent varying magnetic field, the robot can generate different postures and interact with the environment in liquid or on land [1,18,21,22]. Recently, researchers developed robots with various shapes that exhibit versatile modes of movement such as rolling, walking, swimming, jumping, etc. [1,23] Taking rolling as an example, by applying a rotating magnetic field, the magnetic torque is exerted on the robot and therefore the robot rotates if it has a specific magnetization curve. The friction force from the ground or the unbalanced force from the liquid environment propels the robot forward. However, in the biomedical area, it remains challenging to observe real-time information of the robot while endowing it with unique abilities or features.

In this study, we designed a hydrogel-based shape-morphing robot that can be controlled by a rotating magnetic field. The robot is made of magnetic hydrogel materials with a specific defined magnetic profile. When placed in a liquid-enclosed environment, the robot adopts a “C”-shaped posture and rotates forward when the rotating magnetic field is applied (Figure 1a). Additionally, when the robot is stimulated by the near-infrared (NIR) light, the hydrogel network shrinks, leading to its volume change. Under this circumstance, the robot remains capable of propulsion by the external magnetic field, demonstrating its biomedical potential in narrow and complex conditions such as drug delivery in the intestines. To identify the robotic locomotion, ultrasound image technology is applied to efficiently obtain precise posture and position information of the robot. Such dual-stimulated soft magnetic robots with real-time ultrasound imaging provide a promising solution for real-time in vivo monitoring and therapy in biomedical and engineering areas.

## 2. Material and Methods

### 2.1. Fabrication of Dual-Network Hydrogel-Based Millirobot

As shown in Figure 1b, the magnetic/NIR-sensitive millirobot integrates 5 μm magnetic NdFeB particles with dual-network hydrogel. Through a directional magnetization process, the randomly distributed NdFeB powders have magnetic moment with certain direction (Figure 1c), endowing the millirobot with a harmonic magnetization model for C-shaped deformation. Specifically, the dual-network hydrogel material mainly consists of two components: poly(N-isopropylacrylamide) (PNIPAM) and calcium alginate (Ca-Alg) [24]. The first layer network enables the robot to change its volume, while the second layer network ensures that the magnetic particles are dispersed randomly in the network during the fabrication process. To prepare the material, firstly, N-isopropylacrylamide (NIPAM) was mixed in deionized water, then chemical crosslinking agent N, N’-methylenebisacrylamide (BIS, 0.03 mol% of NIPAM) was added to the mixture (Figure 1d). Next, NdFeB powders around 5 μm in diameter (MQFP-B-2007609-089, Magnequench) and sodium alginate (SA) were added. Then, the initiator potassium persulfate was dissolved in the solution, followed by N, N, N’, N’-tetramethyl ethylenediamine. After the solution was well mixed, the prepolymer solution was rapidly poured into a prefabricated cavity. After co-polymerization overnight, the solid polymer was obtained and maintained in 3% (*w*/*v*) CaCl_2_ aqueous solution for about one hour (Figure 1e). The obtained solid polymer was further cut into small rectangular-shaped parts with a length of around 7~10 mm and a width of about 2.5~3 mm. Then these small parts were rolled around a glass rod, and put in a strong magnetic field around 2 T. Since the NdFeB particles have a high residual magnetization, when the external magnetic field (2T) withdrew in the experiment, the magnetization profile of the robot was maintained. It is not re-magnetized in a weak magnetic field (no more than 600 mT). Then the magnetized robots were obtained.

### 2.2. Simulation Methods

To investigate the distribution of the magnetic field and the behavior of the robot in it, simulations shown in Figure 2 were conducted using COMSOL Multiphysics 6.0. For the simulation of the magnetic field, each pair of relatively placed coils were given the same current. For instance, to simulate the rotating magnetic field in the ZY plan, the two Y coils were provided with a sine waveform current, while the two Z coils were given a cosine waveform current.

To simulate the deformation of the robot in a liquid environment, both the magnetic field (with no current) and the fluid–structure interaction field were utilized. In the simulation, the magnetization model of the robot was magnetization, and the central part of the robot was fixed. When the magnetic field was applied, the robot underwent deformation as a result of the magnetic interaction.

## 3. Experiments and Results

### 3.1. Magnetic-Actuated Curve Deformation

The robot was controlled using a custom-designed magnetic system, which consisted of a computer control part, a six-coil system that generated the magnetic field, and two CCD cameras. The coils system comprised of six square coils, and the parameters of each coil were identical, including the number of turns, the inner and outer diameter of the coil, and the wire diameter. The six coils were divided into three pairs, and the distance between each pair of oppositely placed coils was equal to the length of the coil, as shown in Figure 2a. In the experiment, the robot was placed in the central area of the magnetic system.

For a constant current element Idl, the generated magnetic field is proportional to the current I, and the magnetic flux density dB can be calculated as:(1)$$dB=\frac{\mu_{0}}{4\pi }\frac{Idl\times r}{│r{│}^{3}}$$
where r is the radial vector from the current source to the measured point, and $\mu_{0}=4\pi \times 10^{-7}N/A^{2}$ is the magnetic permeability in the vacuum environment. For a square coil, the magnetic field generated on the axis of the coil can be obtained by integrating the magnetic field.

Figure 2b–c shows the simulation results of the magnetic field, where Figure 2b is the simulation meshing of the magnetic coils system. Figure 2c presents the magnetic field in the central working area of the system. The magnetic field at the working area is almost uniform, and as a result, the robot can be assumed to be propelled by the magnetic torque.

Since the robot is magnetized with a specific magnetization curve, the robot deforms and rotates forward under the rotating external magnetic field. The magnetization profile of the obtained robot can be expressed as:(2)$$m=\left(\begin{array}{c} m_{x}\\ m_{y}\\ m_{z} \end{array}\right)=m\left(\begin{array}{c} \sin\left(2\pi *\frac{x}{L}-\pi \right)\\ 0\\ \cos\left(2\pi *\frac{x}{L}-\pi \right) \end{array}\right)$$
where L is the length of the robot, and x is the distance from one end of the robot. The net magnetic moment M of the robot can be expressed as:(3)M=∫mdV

From Equations (2) and (3), the net magnetic moment is theoretically zero when the robot is in an undeformed state. However, when the robot deforms, a net magnetic moment arises.

Since the robot has been magnetized and the magnetization profile is shown in Equation (2), the magnetic moment m of each unit tries to align with the external field when the robot is placed in the magnetic field. As a result, the robot is subject to magnetic torque leading to its deformation or rotation. The magnetic torque is proportional to the magnetic flux density B and the magnetization vector m. For the magnetized hydrogel-based robot, the magnetic torque τ can be calculated by the following formula [25]:(4)$$\tau =m\times B=\left[\begin{array}{ccc} e_{x} & e_{y} & e_{z}\\ m_{x} & m_{y} & m_{z}\\ B_{x} & B_{y} & B_{z} \end{array}\right]$$
where $B={\left(B_{x},\;B_{y},B_{z}\right)}^{T}$, $m={\left(m_{x},\;m_{y},m_{z}\right)}^{T}$, and ${\left(e_{x},\;e_{y},e_{z}\right)}^{T}$ is the identity matrix. Equation (4) can also be rearranged as:(5)$$\tau =\left[\begin{array}{ccc} 0 & B_{z} & -B_{y}\\ -B_{z} & 0 & B_{x}\\ B_{y} & B_{x} & 0 \end{array}\right]\;m$$

Theoretically, when the uniform magnetic field is applied, the direction of all the small units of the robot should be aligned with the external field, and as a result form the shape “C” according to the magnetization profile in Equation (2). Figure 2d demonstrates the numerical simulation results of magnetic robot deformation under the z-axial direction magnetic field. Figure 2d(i) shows the initial state of the robot and Figure 2d(ii)–(iii) show the deforming process of the robot. The simulation result also explains why the robot deforms when actuated by the external magnetic field. Figure 2e provides the experimental results of the C-shaped curve of the millirobot under the magnetic actuation, which agree well with the simulation results.

As described above, when the robot deforms (Figure 2d–e), a net magnetic moment M arises, which can be expressed as [1]:(6)$$M=\int_{0}^{L}RmAds$$
where $R=\left(\begin{array}{ccc} \cos\theta & 0 & -\sin\theta \\ 0 & 1 & 0\\ \sin\theta & 0 & \cos\theta \end{array}\right)$ is the standard y-axis matrix that accounts for the direction change of m due to the deformation of the robot (θ is the rotation angle of each small unit), A is the robot’s cross-sectional area, and ds is the direction along the robot’s length.

The experiment used a uniform rotating magnetic field to propel the robot. For simplification, it can be assumed that the robot forms the shape “C” with the net magnetic moment M first. Then, when the external field rotates, there is an angle between the magnetic moment M and the external field, and the torque T appears, which can be expressed as:(7)T=M×B

When the robot is placed in a rotating magnetic field, there is always an angle between the magnetization direction and the magnetic field, resulting in the continuous rotation and movement of the robot, similar to a wheel.

### 3.2. NIR-Driven Volume Change Capacity

As illustrated in Figure 3a, the thermal responsiveness of PNIPAM in the dual-network hydrogel endows the volume change capacity of the millirobot under the NIR stimulation. PNIPAM exhibits a reversible volume phase transition ability at the lower critical solution temperature (LCST~32 °C) [24]. The temperature-sensitive phase transformation of PNIPAM hydrogel is caused by changes in the hydrophilic/hydrophobic balance of the crosslinking network [26]. In the experiment, the magnetic particles have excellent light–heat conversion capability. Therefore, when the NIR laser is applied, the synthesized hydrogel transfers the laser photon energy into localized heat and exhibits a volumetric shrinkage due to the increasing temperature to the LCST, while the shrinkage recovers reversibly when the laser illumination is off (Appendix A).

Figure 3b,c show the experimental results of the NIR-driven volume change in the synthesized hydrogel patch (length × width: 8 × 7.5 mm). The volumetric shrinkage arrives around 81% with NIR irradiation. After turning off the NIR source, the shrunk volume of the patch recovers gradually since the temperature is lower than the LCST and recovers to around 87% of its original volume during the experiment time. Theoretically, it could recover to the original volume given that the time is long enough.

### 3.3. Dual-Responsive Motility of the Millirobot and Ultrasound Imaging

According to the aforementioned dual responsiveness of the hydrogel millirobot, the integrated motility of the robot is investigated under the hybrid actuation and ultrasound imaging (Figure 4). The magnetized robot was placed in a water environment and controlled under either magnetic or NIR laser control (Appendix A). As shown in Figure 4a, when a rotating magnetic field is applied, the robot deforms into a “C” shape and rotates with the external field, resulting in displacement in the horizontal direction. By adjusting the direction of the rotating magnetic field, the robot can change its moving direction within the container. Upon turning off the magnetic field, the robot stops and recovers to its original shape.

Furthermore, the robot exhibits volume changeability when exposed to the NIR irradiation (Figure 4b). The temperature quickly rises to the LCST after the NIR illumination starts, and the volume of the robot shrinks severely. This provides the robot with high adaptability in narrow environments. Since the robot was controlled in a liquid environment, even when the robot was exposed to the NIR for a long time, the temperature could not reach the curie temperature of NdFeB, which is more than 300 °C. Therefore, the magnetic property of the robot was maintained after NIR illumination and could be controlled using the rotating magnetic field, as shown in Figure 4c. In this way, due to the slow recovery process of the hydrogel material, the shrunk millirobot could be curved and navigated magnetically while maintaining the small volumetric status. The comparison of the average speed before and after NIR illustration is shown in Figure 4d, showing the feasibility of robot locomotion under different volumetric statuses. The dual responsiveness of the proposed millirobot extends the applications of the robot to complex and changing environments such as the intestines.

To observe the robot in real time efficiently and precisely in cases where the robot is placed in an untransparent environment, a diagnostic ultrasound system (MX7, Shenzhen Mindray Bio-Medical Electronics Co., Ltd., Shenzhen, China) was used. The hydrogel robot was placed in a tube full of water in the central part of the magnetic system with the probe of the ultrasound system placed above the tube (Appendix A). Then a rotating magnetic field was applied while the imaging system recorded the real-time posture information including the shape and position of the robot as shown in Figure 4e. This increases the application possibilities of the robot in the biomedical area.

## 4. Conclusions

In this study, we propose a hydrogel-type millirobot with volume-changing capability for magnetic propulsion through a specific magnetization profile. This allows the robot to be propelled by the external rotating magnetic field. Additionally, the robot is equipped with the ability to respond to NIR stimulation, which leads to volume changes, making it suitable for use in complex and confined environments. Furthermore, real-time imaging of the robot can be performed efficiently and precisely through ultrasound technology, highlighting the potential application of this robot in the biomedical field. Overall, compared with previously studied shape-morphing robots that usually have few functions, the proposed millirobot has a unique combination of capabilities that extends its applicability in various conditions.

## Figures and Tables

**Figure 1 micromachines-14-00422-f001:**
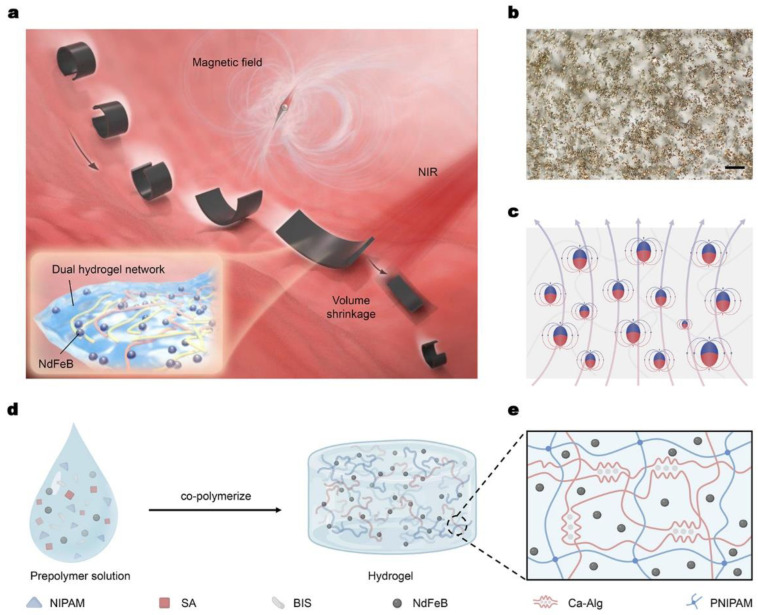
Magnetic/NIR-sensitive hydrogel millirobot for controllable movement and deformation. (**a**) Conceptual schematic of a magnetic/NIR-sensitive hydrogel millirobot in the intestinal environment. The millirobot curves its body and rolls in the magnetic field while shrinking the whole volume in response to the NIR illumination. The inset is the composition of magnetic hydrogel material, including a dual hydrogel network with NdFeB magnetic particles randomly dispersed. (**b**) Optical image of magnetic hydrogel material, and the magnetic particles are dispersed randomly in the material. Scale bar: 100 μm. (**c**) Illustration of magnetic moment distribution after the magnetization process. (**d**) Schematic for the fabrication process of the magnetic/NIR-sensitive hydrogel material. (**e**) Theoretical structure of dual-network hydrogel, composed of crosslinked PNIPAM and Ca-Alg with an interpenetrating polymer network.

**Figure 2 micromachines-14-00422-f002:**
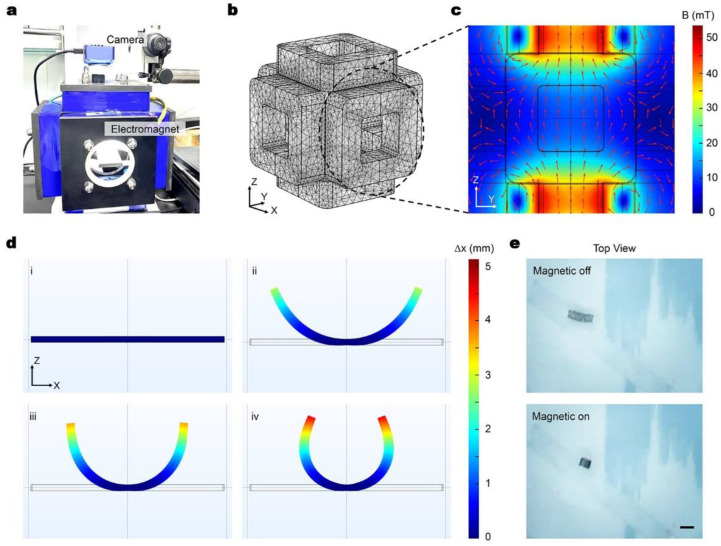
Magnetic actuation of the magnetic/NIR-sensitive hydrogel millirobot. (**a**) Picture of the magnetic generating system setup. (**b**) Mesh refinement of the spatial magnetic system for the simulation. (**c**) Simulation results of the magnetic field in the ZY plan when there is current in the coils. (The red arrows indicate the direction of the magnetic field, while the color map indicates the intensity of the magnetic field.) (**d**) Simulation results when the magnetized robot is placed in the magnetic field, and when the magnetic field is on, the robot will deform (**i**–**iv**). (The color map indicates the displacement of the robot, and the middle part of the robot is fixed.) (**e**) Magnetic-driven deformation of a hydrogel millirobot in a uniform magnetic field. Scale bar: 5 mm.

**Figure 3 micromachines-14-00422-f003:**
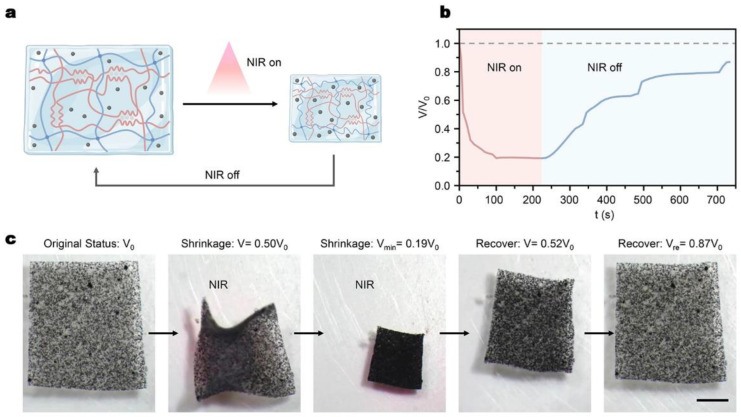
Volume shrinkage of hydrogel material under NIR illumination. (**a**) Schematic for volumetric change in the dual-network hydrogel with NIR illumination on and off. (**b**) Plot of hydrogel volumetric change against 1064 nm NIR treatment time in water. (**c**) Optical images visualizing volume changes in the magnetic hydrogel layer with NIR illumination control. The volume of the robot becomes smaller when the NIR is on, and the volume recovers when the NIR is off. Scale bar: 2 mm.

**Figure 4 micromachines-14-00422-f004:**
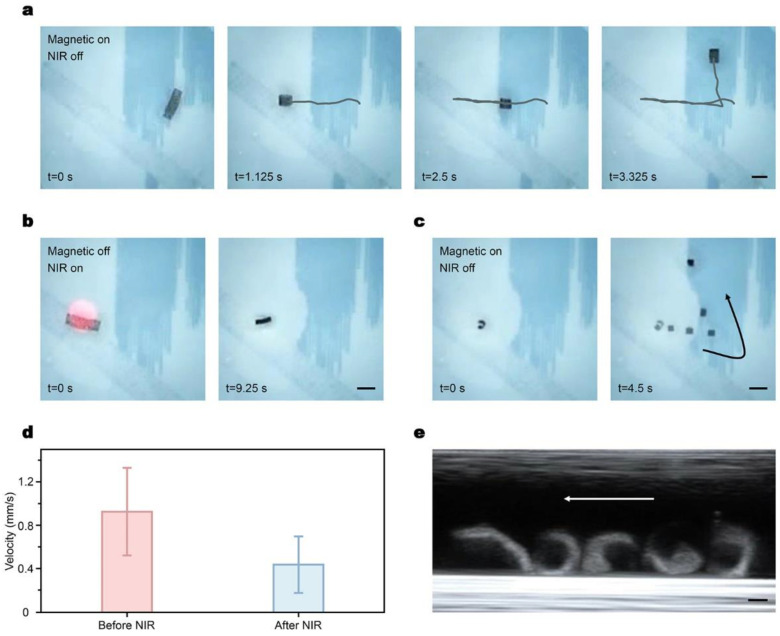
Movement and deformation control of a magnetic/NIR-sensitive millirobot in water under magnetic and NIR control. (**a**) Moving control of the robot in a rotating magnetic field. (**b**) Volume shrinkage of the robot with the NIR irradiation. After irradiation, the volume of the robot changes. (**c**) Moving control of the shrunk robot in a rotating magnetic field. (**d**) Averaged magnetic-driven velocity of the original robot and the shrunk robot after NIR irradiation in the same rotating magnetic field. (**e**) Ultrasound imaging of the robot with magnetic actuation. Scale bar: 5 mm in (**a**)–(**c**) and 2 mm in (**e**).

## Data Availability

Not applicable.

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
