# Peer review of "Ultrasound-Based Real-Time Imaging of Hydrogel-Based Millirobots with Volume Change Capability"

_micromachines, 2023, doi:10.3390/mi14020422_

Round 1
Reviewer 1 Report
The authors proposed a hydrogel-based millirobot that can be controlled with magnetic fields. The work has potential. However, before further publication, several comments need to be addressed.
1. The novelty should be discussed in the introduction. For example, what is the difference or unsolved problem from the previous work?
2. I suggest adding the section "material and methods."
3. Is the proposed material completely new or a modification/improvement of the previous studies? Please cite the relevant studies if it is a modification or adoption of earlier studies.
4. What is the simulation software for finite element simulation? Please describe the detailed software setting in the methods.
5. Are the equations applied for modeling soft robotics for the first time? If not, please also cite the relevant studies. If yes, please highlight the main novelty and add any relevant studies.
6. While discussing the experimental results, please benchmark with the previous studies.
Reviewer 2 Report
1. The preparation process of the robot needs to be described in detail, not only the material preparation process, but also a detailed description of the relevant size parameters of the robot. That is also important for the medical application research of the robot;
2. I recommend that this paper should introduce the driving mechanism of the hydrogel material in detail, how the material realizes deformation and movement through the application of the magnetic field, and its comparative advantages with other magnetic materials;
3. In the research of robot motion performance, this paper needs to add some quantitative data, such as the displacement, angular velocity or deformation;
4. The syntax of this paper needs to be modified uniformly, and should be changed to the past tense.
5.The format of the references needs to be completed, for example, the relevant references need start and end page numbers.
Reviewer 3 Report
Authors present the work done on developing and testing "hydrogel-type miniature robot with volume-changing capability for magnetic propulsion"
The terms soft mini miniature and micro robots are used without much care or justification and used broadly and not in a scientifically adequate way.
The magnetic interaction is presented in a rather simple and basic way. How the movement depends not only on the excitation magnetic field but also on the "robots" is not sufficiently explained.
Some misunderstandings may arise from difficulties the authors may have to express themselves in English language and therefore a revision of the text by a native English speaking scientist is advisable.
Authors exaggerate on the use of images and simulated images and graphs throughout the paper do not explaining sufficiently what they fully represent.
At chapter 2.3 NIR-Driven Volume Change Capacity, the shrinkage and recovery processes are not explained in a sufficiently sound way. Work must to be done at this matter.
The concept of "ultrasound imaging guidance" is a misleading one. There is no evidence the ultrasound imaging was actually used to guide the "micro robot" and certainly not in and automated feedback way.
Round 2
Reviewer 1 Report
All of my comments have been addressed. I have no further comment.
Reviewer 2 Report
The authors have addressed most of my concerns.
Reviewer 3 Report
Authors improved sufficiently their paper that can be accepted in current form